# Tracers for Cardiac Imaging: Targeting the Future of Viable Myocardium

**DOI:** 10.3390/pharmaceutics15051532

**Published:** 2023-05-18

**Authors:** Carmela Nappi, Mariarosaria Panico, Maria Falzarano, Carlo Vallone, Andrea Ponsiglione, Paolo Cutillo, Emilia Zampella, Mario Petretta, Alberto Cuocolo

**Affiliations:** 1Department of Advanced Biomedical Sciences, University of Naples Federico II, 80131 Naples, Italy; 2Institute of Biostructure and Bioimaging, National Council of Research, 80131 Naples, Italy; 3IRCCS SYNLAB SDN, Via Gianturco 113, 80131 Naples, Italy

**Keywords:** ischemic heart disease, viable dysfunctional myocardium, nuclear medicine techniques, tracers, new probes

## Abstract

Ischemic heart disease is the leading cause of mortality worldwide. In this context, myocardial viability is defined as the amount of myocardium that, despite contractile dysfunction, maintains metabolic and electrical function, having the potential for functional enhancement upon revascularization. Recent advances have improved methods to detect myocardial viability. The current paper summarizes the pathophysiological basis of the current methods used to detect myocardial viability in light of the advancements in the development of new radiotracers for cardiac imaging.

## 1. Introduction

Ischemic heart disease is the leading cause of mortality worldwide and is responsible for 8.9 million global annual deaths [1]. Patients with ischemic cardiomyopathy, i.e., those with coronary artery disease and significant left ventricular (LV) systolic dysfunction, with or without heart failure symptoms, have a poor prognosis [2]. These patients may present a significant amount of dysfunctional—although viable—myocardium, with the potential to recover from akinetic or severely hypokinetic status when perfusion improves [3]. In the early era of coronary artery bypass graft surgery, a large body of literature provided evidence regarding the potential benefits of therapeutic intervention according to the size of viable myocardium demonstrating that patients with dysfunctional viable myocardium have improved survival, unlike those with nonviable myocardium for which there is no significant benefit from coronary artery revascularization compared to medical therapy alone [4,5,6,7]. More recent data from the extended STICH (Surgical Treatment for IsChemic Heart failure) study [8] do not support the idea that a long-term benefit of coronary artery bypass graft in patients with ischemic cardiomyopathy can be associated with a pre-surgical assessment of myocardial viability by imaging tests. Although the presence of viable myocardium was indeed associated with improvement in left ventricular systolic function, irrespective of treatment, such improvement was not significantly linked to long-term survival [8]. As limitations, positron emission tomography (PET) data and late gadolinium enhancement by cardiac magnetic resonance (MR) were not included in the STICH study, and myocardial viability was dichotomized and not considered as a continuum. As regard percutaneous coronary intervention, the REVIVED-BCIS2 (Revascularization for Ischemic Ventricular Dysfunction-British Cardiovascular Intervention Society) trial, differently from STICH extended, showed no added benefit of revascularization over optimal medical therapy in patients with severe ischemic cardiomyopathy [9]. It should be also considered that, besides pump failure, sudden death due to ventricular arrhythmias from the border zone between infarcted and viable myocardium may be relevant in determining the poor prognosis of patients with ischemic cardiomyopathy. In other studies, the presence of extensive viability predicts the response to pharmacological treatment [10] and cardiac resynchronization therapy [11]. Hence, myocardial viability may indicate the possibility of obtaining favorable therapeutic results with a range of interventions, other than coronary artery revascularization. Future research into viability testing should consider their application and results more broadly. In particular, viable myocardium should not be considered as a single entity but as a spectrum, including jeopardized, stunned, early hibernation, and advanced hibernation, the revascularization of which may each yield different pathophysiological benefits (Table 1).

Furthermore, for patients with ischemic cardiomyopathy, the question is not whether they have viable myocardium but how much they have of each type, where it is, whether it is likely to recover and how long this may take. Central to this is the need for multidisciplinary collaboration to integrate, even with the help of machine learning, clinical information with the myocardial and coronary substrate and imaging data. Noninvasive methods for detecting viability, including echocardiography, cardiac MR and perfusion computed tomography (CT), and nuclear medicine techniques such as single-photon emission CT (SPECT) and positron emission tomography (PET) are rapidly evolving [12,13,14,15]. Of note, these methods differ in the pathophysiological principles used for the assessment of myocardial viability, such as preserved membrane and mitochondrial function, presence and degree of contractile reserve, myocardial metabolism, or absence of myocardial scar (Table 2).

Echocardiography is the oldest noninvasive method to study heart and its potential usefulness for assessing viability (mostly by dobutamine stress test) has been demonstrated [16]. Despite the advantages (lower cost, availability and the lack of ionizing radiation or renal toxicity), the procedure carries many limitations such as technical difficulties in acquiring images in patients with poor acoustic windows and low agreement and reliability, especially in case of limited image quality [17]. In addition, myocardial strain analysis has been proposed as a tool to better identify the presence of viable myocardium [18]. Cardiac MR is one of the most accurate techniques in the arsenal. Using contrast-delayed enhancement, it is possible not only to assess the severity and extension of infarcted myocardium but also to investigate myocardial viability [19]. The limited availability of MR technologies and the restricted possibilities to evaluate patients with implantable devices are the principal limitations of this method. Recently, new technological MR-compatible devices (pacemakers and defibrillators) are being increasingly used, but it should be stressed that they may cause significant imaging artifacts [20]. CT scan (by different protocols) shows high accuracy in myocardium trials and infarct detection [21]. An emerging technique, dual-energy CT, seems to be very promising for myocardial studies and it seems to be also useful to investigate myocardial viability for future promising applications [22,23]. Nuclear medicine procedures by PET or SPECT are widely used to assess cardiovascular risk and prognosis and evaluation of therapy in patients with coronary artery disease, and are also useful for assessment of myocardial viability [24]. SPECT imaging by ^201^Tl and ^99m^Tc-sestamibi has a respectable indication to evaluate myocardial viability [25]. Its use has two important advantages: good availability and few contraindications [26]. PET is useful both for perfusion studies by using ^13^N-ammonia, ^82^Rb, or ^15^O-water and for myocardial viability studies by using radiolabeled ^18^F-2-deoxy-2-fluoro-D-glucose (FDG) (Figure 1). ^18^F-FDG PET is the method with the best sensitivity and specificity to investigate myocardial viability and predict functional myocardial recovery after revascularization [27]. Currently, nuclear medicine is on the rise thanks to the development of new radiotracers with great potential to be considered new molecular probes in the near future. The aim of the present review is to summarize the state-of-the-art of available tracers for the evaluation of viable myocardium and to provide an overview of future prospectives.

## 2. Pathophysiological Bases of Current Tracers

Recurrent episodes of transient post-ischemic dysfunction (myocardial stunning) due to ischemia and reperfusion not so severe or so long as to cause myocellular necrosis as well as a chronically reduced perfusion can initiate a protective state of myocardial down-regulation called “myocardial hibernation”. This is an advanced state of ischemic dysfunction, histologically characterized by myocellular dedifferentiation, change in gene expression, losses of the contractile proteins and alterations of myocardial metabolism, which starts to prefer the use of glucose instead of free fatty acid, with energy production mainly depending on anaerobic glycolysis [13,14]. These changes in potential molecular targets may be used to study different pathophysiological aspects of myocardium injury [15]. Several techniques, all with advantages and limitations, demonstrated the potential usefulness to study myocardial viability, none better than nuclear medicine procedures, including the use of different tracers for PET and SPECT application (Table 3), with the highest sensitivity observed for PET by using dedicated protocols with the administration of ^18^F-FDG [27].

### 2.1. PET Tracers

The clinical applicability of ^18^F-FDG tracer is based on intracellular trapping after phosphorylation because it produces a substrate that is not useful for metabolism but is helpful to detect metabolic cellular changes during ischemia. The imaging evaluation is realized by a semiquantitative method based on comparatively regional uptake of ^18^F-FDG [27]. To investigate myocardial viability, a combined evaluation of perfusion and metabolism is required [28]. While ^18^F-FDG tracer can be used to investigate metabolism, perfusion imaging (performed in the rest phase, and the stress phase too but only where necessary) is commonly realized using ^13^N ammonia or ^82^Rb tracers. In hypoperfused and dysfunctional myocardium, it is possible to identify cell survival and viability in the presence of ^18^F-FDG uptake in the same territory (flow-metabolism mismatch) or not viable tissue in case of not-detectable ^18^F-FDG uptake (flow-metabolism matched defect) [29]. Representative examples of patients with different perfusion-metabolism patterns by combined ^82^Rb/^18^F-FDG PET imaging are depicted in Figure 2 and Figure 3.

Of note, a potential further pattern can be observed with reversed perfusion-metabolism mismatch when there is a reduction in ^18^F-FDG activity in the septum with normal perfusion due to a left bundle branch block [30]. The combined evaluation of metabolism and perfusion is the preferred method to evaluate myocardial viability [31]. However, ^18^F-FDG administration requires patient preparation (patient fasting for at least 6 h, oral or intravenous glucose load) to stimulate endogenous insulin release and to minimize the variability in substrate setting between patients. After glucose load, insulin can be administered as needed [28]. Many studies have reported the good performance of this method [27,32,33]. A meta-analysis showed a mean sensitivity of 93% and specificity of 58% to predict myocardium recovery and improvement in left ventricular ejection fraction after revascularization [27]. However, it should be considered that the accuracy decreases in diabetic patients, requiring a more complex preparation protocol [34]. Conversely, the evaluation of oxidative metabolism can be used to identify viable myocardium. Considering the possibility to have an early oxidative metabolism after thrombolysis, it has been noted that the left ventricular functional recovery can be related to the improvement in oxidative metabolism, this condition can be investigated by ^11^C-acetate PET tracer. The reduction in this tracer uptake can be related to a necrotic area in patients with recent myocardial infarction [35]. Free fatty acids are the major energy substrate for healthy cardiac muscle. Thus, palmitate labeled with ^11^C has been considered a promising candidate to investigate myocardial viability by PET scan. Its uptake and its metabolism can be related to perfusion, oxygenation levels and neurohumoral environment fitting for cardiac metabolism studies. Nevertheless, the difficult washout limited the routine use of this tracer in clinical practice [35]. To investigate simultaneously perfusion and metabolism using one tracer and reducing both radiation exposure and imaging time, the utilization of ^15^O-water has been also proposed considering the concept that the injured myocardial regions have a reduced capability to provide rapid water exchange. This method, interesting but hardly feasible, is reserved for medical centers equipped with a cyclotron due to the very short ^15^O radionuclide half-life of 120 s. Regarding this tracer, an excellent study was carried out by Grönman et al. [36]. In a pig model, they demonstrated that the quantification (based on a single-compartment model) of perfusion parameters like myocardial blood flow, perfusion tissue fraction and perfusion tissue index can provide useful data for myocardial viability assessment. However, other studies are elsewhere required.

### 2.2. SPECT Tracers

^201^Tl can be used to differentiate between viable and nonviable myocardium [37,38]. This is an analog of potassium, which is present in myocytes and absent in scar tissue; its high first-pass extraction is proportional to coronary blood flow. The initial myocardial activity of ^201^Tl is driven by the coronary blood flow state at rest whereas the later uptake over the next 4–24 h is determined by the tracer redistribution, depending on cellular membrane integrity. Hibernation may appear as a myocardial perfusion defect on initial imaging due to reduced coronary blood flow at rest but normalizes on the delayed imaging (with or without ^201^Tl reinjection) from redistribution of the tracer. The sensitivity of viability detection by ^201^Tl imaging may increase in the late (24 h) reinjection/redistribution protocol compared to the 4 h early redistribution protocol. The radiopharmaceutical activity of ^201^Tl in rest/redistribution imaging is approximately 3 mCi with a corresponding radiation effective dose of 10–15 mSv. The uptake into the myocyte happens by the Na^+^/K^+^-adenosine triphosphate transport system and by facilitative diffusion, too [39]. ^201^Tl retention requires a whole sarcolemma. The redistribution starts about 15 to 20 min after injection. Watching thallium uptake, retention, and redistribution, using the two main protocols (rest- and stress-redistribution) for myocardial perfusion, sarcolemma membrane function and myocyte metabolic activity to identify myocardial viability can be investigated [40]. For the evaluation of myocardial ischemia, a stress-redistribution protocol is required with ^201^Tl injected just before peak exercise or at peak pharmacologic vasodilatation. Stress imaging is obtained 15 min after administration. If a fixed perfusion defect is noted, the redistribution images are obtained after 24 h from tracer injection [41]. A meta-analysis reported a mean sensitivity of 87% and a specificity of 54% of ^201^Tl imaging to predict recovery of left ventricular function after coronary revascularization [42]. ^99m^Tc-labeled tracers (sestamibi and tetrofosmin), are commonly used in the evaluation of ischemia; they can also be used in the same protocols to assess myocardial viability. ^99m^Tc is a lipophilic substrate, it enters in myocytes by passive diffusion. Differently from ^201^Tl, the redistribution of ^99m^Tc is limited. Uptake and retention of ^99m^Tc require intact mitochondrial and sarcolemma membranes, which reflect myocyte viability. ^99m^Tc sestamibi has been shown to be comparable to ^201^Tl in its ability to predict regional recovery of function following revascularization [42,43]. However, it should be taken into account that the common ^99m^Tc-labeled tracers cannot be considered ideal tracers given their relatively low first-pass extraction fraction and high liver absorption, although they boast low lung extraction. Conversely, the advent of ultrafast high-sensitivity gamma cameras paved the way to evaluate potential novel probes with high initial absorption of the heart, combined with longer myocardial retention and high heart/background ratios such as ^99m^Tc-Teboroxime [44]. It has been demonstrated that the addition of a short-acting nitrate prior to administration of ^99m^Tc-sestamibi may expand sensitivity [45,46]. The performance of sestamibi SPECT can be considered comparable to that of ^201^Tl to detect reversible left ventricular dysfunction [47]. Moreover, in patients with myocardial infarction and chronic reduction of left ventricular function, the diagnostic results carried out by the quantitative rest tetrofosmin analysis are similar to results from ^201^Tl and sestamibi imaging [48]. Knowledge of beta-methyl-iodophenyl-pentadecanoic acid (BMIPP) dates back several decades. In particular, a great effort has been made by Japanese researchers to provide the basis for its possible employment as a novel diagnostic and prognostic probe labeled with ^123^I to study acute myocardial infarction [49,50,51]. Combined perfusion studies and fatty acid metabolism evaluation by ^123^I-BMIPP imaging may produce three different situations: defects of both perfusion and fatty acid metabolism imaging, representing scar or nonviable tissue; lower ^123^I-BMIPP uptake compared to perfusion tracers’ distribution implicating metabolically damaged but viable myocardium; normal uptake of both perfusion and metabolic tracers meaning healthy myocardium. Identification of these perfusion-metabolism correlations helps to detect viable and nonviable myocardium [52]. However, the accuracy of combined SPECT imaging by using ^123^I-BMIPP and ^99m^Tc-labeled perfusion tracers is lower than the combined imaging by perfusion SPECT and ^18^F-FDG PET imaging studies. Thus, the latter method is preferred [53].

## 3. The Way to Future Imaging

The strength of PET as an imaging technique relies on the versatility of positron-emitting radionuclides that can be integrated into important biochemical molecules. Not only can the distribution of these molecules be imaged, but their uptake can be quantified. In this way, it is possible to assess myocardial perfusion, glucose utilization, fatty acid uptake and oxidation, oxygen consumption, contractile function and presynaptic and postsynaptic neuronal activity [54]. Using specific radiotracers, targets of different pathophysiological steps, the state of advancement of ischemic dysfunction may be investigated. This is the aim of the recent advancement in radiotracer development. Novel molecular radioligands have been developed to study specific cellular components of the inflammatory response after myocardial injury [55,56,57,58]. In particular, the remodeling tissue can be investigated using specific tracers as indirect probes of myocardial suffering, identifying different phases of myocardial disease onset, such as the identification of apoptosis, macrophage presence, or fibroblast activation. In this context, a number of novel molecules have been developed and proposed on the research ground. The 1,4,7,10-tetraazacyclododecane-1,4,7,10-tetraacetic acid-extracellular loop 1 inverso (DOTA-ECL1i) labeled with ^68^Ga is a useful peptide for PET scan, with a selectively increased binding in presence of a C-C chemokine receptor type 2 (CCR2) ligand [58,59]. Cells expressing CCR2, like monocyte, T cells, or B cells are present in large numbers in inflammatory states following an acute event such as myocardial infarction. Therefore, ^68^Ga-DOTA-ECL1i has been recently investigated to track the recruitment, accumulation and resolution of CCR2+ monocytes and macrophages in the injured myocardium in a mouse model, demonstrating its capacity to identify inflammation involving both the infarct and peri-infarct areas [60]. Other studies on its use are required for clinical translation. It also should be taken into account that in response to an injury (ischemic or not), myofibroblasts produce the fibroblast activation protein (FAP) [61]. There are many inputs for fibrosis, such as myocyte death or mechanical stimuli like pressure, volume overload and neurohormonal activation [62]. The persistence of injury supports progressive fibrogenesis over time [63]. Unfortunately, this target stage is the last step of disease. Therefore, new biomarkers of fibroblast activation to investigate the early stages of the pathology are needed, and increasing attention has been focused on ^68^Ga-FAP inhibitor development [64] because of its capacity to link up with FAP, produced by fibroblast, in an inflammatory status [65]. A high level of FAP in myofibroblasts has been reported in infarcted hearts [66]. Fibroblast activation evaluation and monitoring are possible by ^68^Ga-FAP inhibitor, and it can be used to investigate myocardial conditions associated with fibroblast activation [66,67]. Recently, mitochondrial membrane integrity has been proposed as a useful candidate to study the condition in myocardial tissue. The opportunity to quantify the membrane potential allows a direct comparison between subjects and it would be particularly relevant to study the state of advancement of ischemic dysfunction [68]. Mitochondria produce most (90%) cellular adenosine triphosphate (ATP) by oxidative phosphorylation [69]. The mitochondrial electron transport chain converts nutrients into energy. Thus, the mitochondrial membrane potential is needed for the conversion of adenosine diphosphate to ATP [70]. A change in the mitochondrial membrane potential produces not only the overproduction of reactive oxygen species but also lower ATP production and an increase in pathological conditions [70]. Alpert, et al. [71] introduced a method to perform in vivo imaging with PET/CT to measure and map the total membrane potential of cells by a cationic lipophilic tracer, the ^18^F-labeled tetraphenylphosphonium. This method was used for the quantitative mapping of total membrane potential in swine myocardial cells [71]. In a follow-up study, Pelletier-Galarneau et al. [72] confirmed the strength of this methodology and the possibility to apply the technique also in humans demonstrating that in 13 healthy people cellular membrane potential and mitochondrial membrane potential were in excellent agreement with the prior evaluation in vitro [72]. Of note, it has been demonstrated that a strong correlation between changes in the density of adrenergic receptors and viable dysfunctional myocardium exists [73]. Based on this evidence, data studies suggest an interesting application of SPECT ^123^I-meta-iodo benzyl-guanidine (^123^I-MIBG) and PET studies by using different innervation tracers such as ^18^F-Flubrobenguane (FBBG, also known as ^18^F-LMI-1195), ^18^F meta-fluorobenzylguanidine (^18^F-MFBG), or ^11^C-hydroxyephedrine (^11^C-HED) [74,75]. An example of a patient with non-viable myocardium and defect of innervation in the same territory is shown in Figure 4. While the use of ^23^I-MIBG imaging may count on a robust body of literature including large clinical trials, and it boasts widespread availability of the tracer around the world, the lower resolution of gamma cameras as compared with PET scan may represent a limitation. At the same time, ^11^C-HED PET/CT imaging benefits from the implied advantages of the scan, such as higher temporal and spatial resolution, making possible absolute quantification. However, the short radionuclide half-life requires on-site production of the compound. Yet, the utilization of ^18^F-labeled sympathetic molecules may overcome this limitation [76]. This may be the dawn of a technique that could be useful to evaluate different patients and their different responses to therapy. Surprising data may also further come from the use of tracers with primary oncological scope to evaluate not only myocardial viability but also cardiac metastatic involvement. In this context, a potential utilization of prostate-specific membrane antigen labeled with ^68^Ga has been reported [77]. This is consistent with the concept that prostate-specific membrane antigen is upregulated on the endothelial cells of the neovasculature of a variety of other solid tumors [78]. Table 4 provides a list of ongoing clinical trials using cardiovascular imaging tracers and Table 5 illustrates the advantages and limitations of cardiovascular imaging tracers.

## 4. Conclusions

In the time of personalized medicine, the identification of methods that break down technical and clinical limitations is the new challenge. In the context of research for innovation in cardiovascular imaging, a potential breakthrough could be represented by new tracer development to assess myocardial viable tissue with the ability not only to improve diagnostic performances but also to refine the current knowledge on myocardial viability physiological and pathophysiological patterns.

## Figures and Tables

**Figure 1 pharmaceutics-15-01532-f001:**
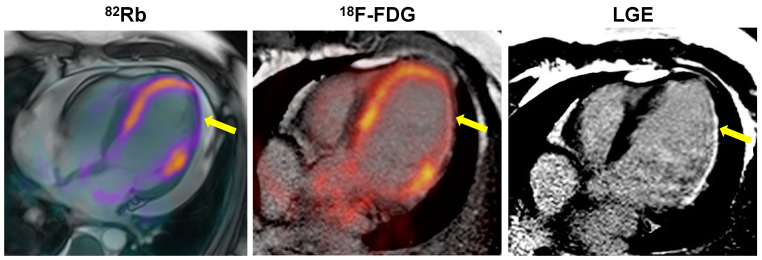
Patient with a large area (yellow arrows) of hypoperfused (^82^Rb PET/MR) and nonviable (^18^F-FDG PET/CT) myocardium in the inferolateral wall of the left ventricle corresponding to an MR pattern of late gadolinium enhancement (LGE) in the same region.

**Figure 2 pharmaceutics-15-01532-f002:**
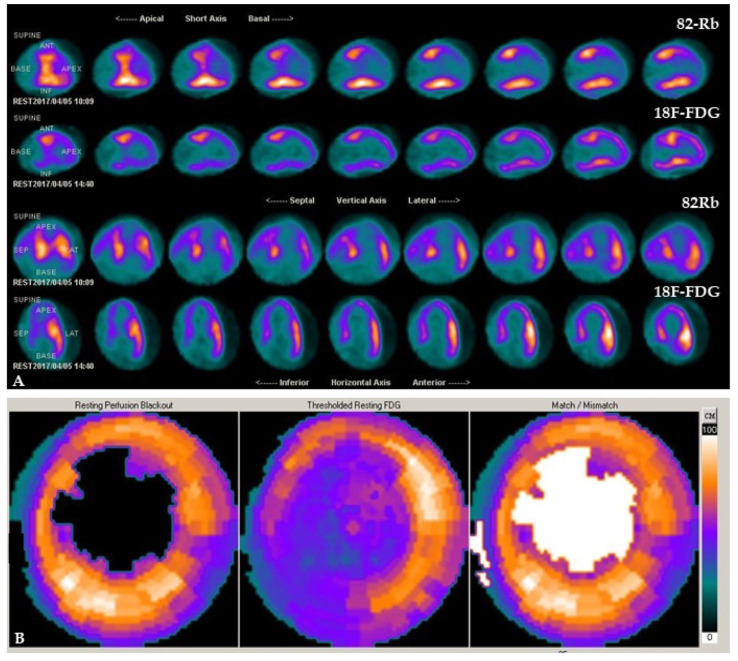
Patient with severe hypoperfusion and preserved metabolic activity in the apex and the apical segment of the anteroseptal region of the left ventricle by combined ^82^Rb/^18^F-FDG PET/CT cardiac imaging. Slice review (**A**) and polar maps (**B**) indicate a mismatch pattern that suggests the presence of hypoperfused but viable myocardium.

**Figure 3 pharmaceutics-15-01532-f003:**
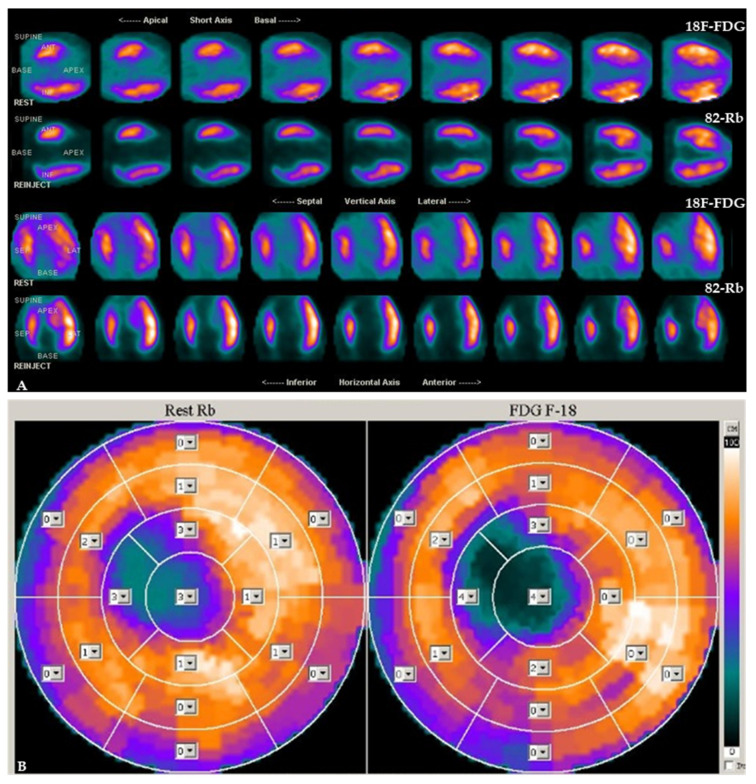
Patient with absent perfusion and no evidence of metabolic activity in the apex and the apical segment of the anteroseptal region of the left ventricle by combined ^82^Rb/^18^F-FDG PET/CT cardiac imaging. Slice review (**A**) and polar maps (**B**) indicate a match pattern that suggests the presence of necrotic myocardium. Analysis of tracer uptake is performed in 17 myocardial segments by use of a 5-point scoring system (from 0, normal uptake to 4, absence of detectable tracer uptake).

**Figure 4 pharmaceutics-15-01532-f004:**
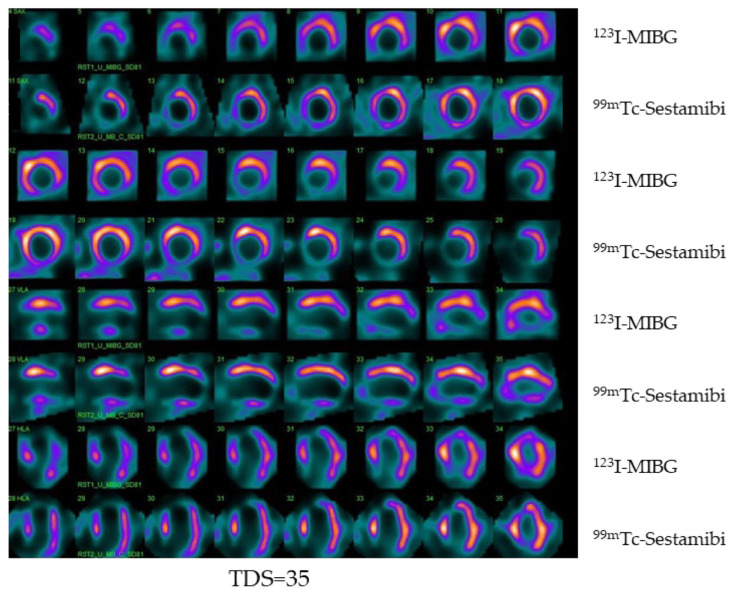
Example of dual isotope ^123^I-MIBG/^99m^Tc-sestamibi imaging obtained with CZT camera in a patient with hypertension and dyslipidemia and heart failure. Tomographic images (first row ^123^I-MIBG, second row ^99m^Tc-sestamibi) demonstrating innervation and perfusion defect on the inferior wall of the left ventricle with a total defect score (TDS) of 35.

**Table 1 pharmaceutics-15-01532-t001:** Different myocardial states.

	Function	Perfusion	Metabolism	Inotropic Response	Functional Recovery
Stunned (viable)	Reduced	Normal	Yes	Yes	Yes
Hibernating (viable)	Reduced	Reduced	Yes	Yes	Yes
Necrotic (non-viable)	Reduced	Reduced	No	No	No

**Table 2 pharmaceutics-15-01532-t002:** Noninvasive imaging methods to assess myocardial viability.

	Measure	Viability Marker
SPECT	^201^Tl or ^99m^Tc uptake	Myocyte membrane integrity
PET	^18^F-FDG uptake	Glucose metabolism
Stress echocardiography	Inotropic stimulation	Contractile reserve
Cardiac MR	Late gadolinium enhancement	Extracellular volume

SPECT: single-photon emission computed tomography, PET: positron emission tomography, FDG: 2-deoxy-2-fluoro-D-glucose, MR: magnetic resonance.

**Table 3 pharmaceutics-15-01532-t003:** Characteristics of radiotracers for imaging myocardial viability.

Tracer	Half-Life	Pathway	Emission	Energy (MeV)
^18^F-FDG	110 min	Hexokinase/glucose metabolism	β+	0.633
^11^C-acetate	20 min	Krebs cycle/free fatty acid metabolism	β+,	0.961
^15^O-water	2 min	Passive diffusion/blood flow	β+	0.019
^68^Ga-DOTA-ECL1i	68 min	CCR type 2/inflammatory response to injury	β+	1.899
^68^Ga-FAPI	68 min	FAP/inflammatory response to injury	β+	1.899
^18^F-tetraphenylphosphonium	110 min	Mitochondria/mitochondrial membrane integrity	β+	0.633
^123^I-MIBG	13.2 h	Distribution and integrity of adrenergic nerve endings	γ	0.159
^18^F-FBBG (or ^18^F-LMI-1195)	110 min	Distribution and integrity of adrenergic nerve endings	β+	0.633
^18^F-MFBG	110 min	Distribution and integrity of adrenergic nerve endings	β+	0.633
^11^C-HED	20 min	Distribution and integrity of adrenergic nerve endings	β+	0.961
^201^Tl	72.9 h	Na^+^/K^+^ pump/blood flow	γ	0.135, 0.167
^123^I-BMIPP	13.2 h	Krebs cycle/free fatty acid metabolism	γ	0.159
^99m^Tc-Sestamibi	6 h	Mitochondria and cytosol proteins in myocytes	γ	0.140
^99m^Tc-Tetrofosmin	6 h	Mitochondria and cytosol proteins in myocytes	γ	0.140

FDG: 2-deoxy-2-fluoro-D-glucose, DOTA-ECL1i: 1,4,7,10-tetraazacyclododecane-1,4,7,10-tetraacetic acid-extracellular loop 1 inverso, CCR: C-C chemokine receptor, FAPI: fibroblast activation protein inhibitor, MIBG: metaiodobenzylguanidine, FBBG: flubrobenguane, MFBG: meta-fluorobenzylguanidine, HED: hydroxyephedrine, BMIPP, beta-methyl-iodophenyl-pentadecanoic acid.

**Table 4 pharmaceutics-15-01532-t004:** Current ongoing clinical trials using cardiovascular imaging perfusion and viability tracers.

Project Title	Sponsors	Study Type	Aim	Status
Development and Translation of Generator-Produced PET Tracer for Myocardial Perfusion Imaging-Dosimetry Group (GALMYDAR)	Washington University School of Medicine, USA	Interventional	To evaluate dosimetry, biodistribution, safety and imaging characteristics following a single ^68^Ga-Galmydar injection in normal healthy volunteers	Recruiting completed
68 Ga-NODAGA-E[c(RGDγK)]2: Positron Emission Tomography Tracer for Imaging of Myocardial Angiogenesis	Rigshospitalet, Denmark	Interventional	To examine the expression of αvβ3 integrin using a novel radiotracer in patients with myocardial infarction and investigate if it is a suitable tool for predicting myocardial recovery and prognosis	Recruiting completed
Cardiac FDG PET Viability Registry (CADRE)	Ottawa Heart Institute Research Corporation, Canada	Observational	To evaluate the utility of FDG PET imaging in the decision-making process for patients with poor left ventricular function who may be candidates for revascularization and to study the downstream effect of the clinical management decisions	Recruiting
Open-Label, Exploratory, Phase 1/2 Scintigraphy Study Evaluating ^18^F-mFBG for Imaging Myocardial Sympathetic Innervation in Subjects Without and With Heart Disease	Innervate Radiopharmaceuticals LLC, USA	Interventional	To observe the positron-emitting radiopharmaceutical ^18^F-mFBG as an imaging agent for quantification of myocardial sympathetic innervation	Recruiting
Phase 3, Multicenter, Open Label Study to Confirm the Diagnostic Potential of Intravenously Administered ^15^O-H_2_O to Identify Coronary Artery Disease During Pharmacological Stress and Resting Conditions Using PET Imaging (RAPID-WATER-FLOW)	MedTrace Pharma A/S, Denmark	Interventional	To evaluate the sensitivity and specificity of the ^15^O-H_2_O PET study using the truth-standard of ICA with FFR or CCTA	Recruiting

**Table 5 pharmaceutics-15-01532-t005:** Advantages and limitations of cardiovascular imaging tracers.

Tracer	Advantages	Limitations
^18^F-FDG	Long radionuclide half-life allowing delivery; high temporal and spatial resolution of equipment; robust evidence	Metabolic compensation needed; perfusion study required for a combined evaluation
^11^C-acetate	Single-tracer technique; minimal metabolic dependence	On-site cyclotron required
^15^O-water	High and linear tracer; uptake rate into myocardium; high temporal and spatial resolution of equipment	On-site cyclotron required; technical demanding protocols
^68^Ga-DOTA-ECL1i	Commercially available ^68^Ge/^68^Ga generator for multiple daily studies; rapid clearance; low liver retention	Very limited data available; low specificity
^68^Ga-FAPI	Commercially available ^68^Ge/^68^Ga generator for multiple daily studies; high abnormal/normal uptake ratio	Limited data available; low specificity
^18^F-tetraphenylphosphonium	Long radionuclide half-life allowing delivery; High temporal and spatial resolution of equipment; first voltage non-invasive probe	Limited data available; no gold standard method as reference; high distribution heterogeneity
^123^I-MIBG	Robust evidence; optimal storage in neuronal vesicles; highly specific tracer; high heart-to-background ratios with clear cardiac images	Low resolution of equipment; standardization protocols still required
^18^F-FBBG (or ^18^F-LMI-1195)	Simple radiolabeling; procedure for commercial use; high heart-to-background ratios with clear cardiac images; high temporal and spatial resolution of equipment	Limited data available
^18^F-MFBG	Optimal storage in neuronal vesicles; highly specific tracer; high heart-to-background ratios with clear cardiac images; high temporal and spatial resolution of equipment	Limited data available
^11^C-HED	Robust evidence; highly specific tracer; high heart-to-background ratios with clear cardiac images high temporal and spatial resolution of equipment	On-site cyclotron required; delayed scans for turnover assessment not feasible due to low radionuclide half-life; high lipophilicity with potential tracer loss across lipid membranes
^201^Tl	Tissue concentration proportional to flow; potential evaluation of perfusion and viability	Low resolution of equipment; dosimetric issues
^123^I-BMIPP	Primary energy cardiac source tracer; high specificity	Low resolution of equipment; low sensitivity
^99m^Tc-Sestamibi and ^99m^Tc-Tetrofosmin	Short radionuclide half-life with a feasible dosimetric profile; myocardial uptake proportional to the integrity of membrane with high accuracy	Low resolution of equipment; low first-pass extraction fraction and high liver absorption

FDG: 2-deoxy-2-fluoro-D-glucose, DOTA-ECL1i: 1,4,7,10-tetraazacyclododecane-1,4,7,10-tetraacetic acid-extracellular loop 1 inverso, CCR: C-C chemokine receptor, FAPI: fibroblast activation protein inhibitor, MIBG: metaiodobenzylguanidine, FBBG: flubrobenguane, MFBG: meta-fluorobenzylguanidine, HED: hydroxyephedrine, BMIPP, beta-methyl-iodophenyl-pentadecanoic acid.

## Data Availability

Data sharing not applicable.

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
