# Peer review of "Tracers for Cardiac Imaging: Targeting the Future of Viable Myocardium"

_pharmaceutics, 2023, doi:10.3390/pharmaceutics15051532_

Round 1

Reviewer 1 Report

Line 15 enhancement  upon  enhancement upon (double space??)

In rule of this journal, do you need to have a space between the number and %?

Table footnote font is the same as the text, so it's a little hard to read. 

Figure 1 will be better understood if the parts are shown.

It is difficult for anyone other than an expert in the field to understand.

Table 3 should be easier to read with a little more effort (lower font, line breaks, narrower width).

Is it possible to increase the resolution of Figure 2 a little more?

It is necessary to clearly indicate this figure from which direction of the body is viewed.

In Figure 3, the characters in the figure are distorted, making it difficult to understand. 

Line 36-40. Authors should write this part a little more unique.

These sentences are the same as the source.

Line 491 123I  123I

Is the nuclide notation consistent with the source? Please correct the whole place where it is not a superscript character. (Especially in reference section)

Line 521 ”89(,” is what??

Line 345-351: Shouldn't the font here be aligned to the smaller one?

Certainly, the conclusion is as the authors say, but since they have taken the trouble to summarize it as a review, do the authors have any ideas that might be a breakthrough?

Author Response

Line 15 enhancement upon → enhancement upon (double space??)

Response: Mistyping has been edited

In rule of this journal, do you need to have a space between the number and %?

Response: In the manuscript there are no spaces between the number and %.

Table footnote font is the same as the text, so it's a little hard to read.

Response: The font size has been made bigger.

Figure 1 will be better understood if the parts are shown.

It is difficult for anyone other than an expert in the field to understand.

Response: To make it easier to read, some arrows have been added in the figure.

Table 3 should be easier to read with a little more effort (lower font, line breaks, narrower width).

Is it possible to increase the resolution of Figure 2 a little more?

It is necessary to clearly indicate this figure from which direction of the body is viewed.

Response: Figure 2 resolution has been increased. We also made understandable the direction of slices.

In Figure 3, the characters in the figure are distorted, making it difficult to understand.

Response: Figure 3 resolution has been increased. We also made understandable the direction of slices.

Line 36-40. Authors should write this part a little more unique.

These sentences are the same as the source.

Response: The paragraph has been edited according to reviewer suggestions.

Line 491 123I → 123I

Is the nuclide notation consistent with the source? Please correct the whole place where it is not a superscript character. (Especially in reference section).

Response: The reference section has been edited.

Line 521 ”89(,” is what??

Response: This mistype has been corrected.

Line 345-351: Shouldn't the font here be aligned to the smaller one?

Response: The font has been reduced.

Certainly, the conclusion is as the authors say, but since they have taken the trouble to summarize it as a review, do the authors have any ideas that might be a breakthrough?

Response: The conclusion paragraph has been edited to make clearer that the development of new tracers can be the breakthrough.

Reviewer 2 Report

The authors provide a comprehensive review of tracers for Cardiac Imaging.

This is a well-written work. Would suggest as a minor comment to add some context with respect to cancer patients with cardiac issues or/and cardiac metastases (although rare) and how different PSMA imaging modalities might interfere with tumor neovasculature.

Author Response

Response: A comment has been added at the end of discussion and two more references (#77,#78) have been quoted.

Reviewer 3 Report

It is well organized review. But three complements are needed.

1. The pros and cons of the current tracers should be provided in a separate table or merged into table 3.

2. A separate table should be provided including the known weaknesses and improvements of the tracers mentioned in section 3, 'The Way to Future Imaging'.

3. There are some clinical trials for the recently renewed (improved) tracers. It would be suggested to list the trials as a separate section (e.g. https://clinicaltrials.gov/ct2/show/NCT05280782; https://clinicaltrials.gov/ct2/show/NCT03445884; etc.).

Author Response

  1. The pros and cons of the current tracers should be provided in a separate table or merged into table 3.

Response: A separate table has been provided (Table 5) listing pros and cons of current tracers.

  1. A separate table should be provided including the known weaknesses and improvements of the tracers mentioned in section 3, 'The Way to Future Imaging'.

Response: A separate table has been provided (Table 5) listing pros and cons of mentioned tracers.

  1. There are some clinical trials for the recently renewed (improved) tracers. It would be suggested to list the trials as a separate section (e.g., https://clinicaltrials.gov/ct2/show/NCT05280782; https://clinicaltrials.gov/ct2/show/NCT03445884; etc.).

Response: A table summarizing the ongoing clinical trials has been added.

Reviewer 4 Report

The review paper on radiotracers for cardiac imaging, submitted by Carmela Nappi et al., is well-organized, well-written and covers the most important developments in this field. However, excessive clinical details are given in some parts of the manuscript. By contrast, a more comprehensive comparison of the performance of the different radiotracers is missing in some cases. I will be happy to recommend the publication of the manuscript after the authors address the following issues:

i)                Page 2, Line 37: Define the CABG abbreviation.

ii)              Page 3, Line 77: Write “fluorodeoxyglucose” as 2-deoxy-2-fluoro-D-glucose” all along the text.

iii)             Page 4, entries of Table 4: Replace “Half-life” by "Radionuclide Half-life”, to avoid confusion with the biological half-life of the radiopharmaceutical.

iv)             Page 4, Table 4: For 11C-acetate and 11C-HED, uniformize the value of positron energy (either 0.961 MeV or 0.96 MeV).

v)              Page 5, Table 4: Rewrite “99mSestamibi” and “99mTetrofosmin” as “99mTc-Sestamibi” and “99mTc-Tetrofosmin”, respectively.

vi)             Page 5, Line 147: I believe that the authors wanted to mean “where” instead of “were”. Please, correct.

vii)            Page 6 and 7, Figures 2 and 3: The resolution of the images needs to be improved.

viii)           Page 8, Line 184: Correct “11C” to “11C”.

ix)         Page 8, Line 190: Clarify how 15O-water can assess myocardium metabolism.

x)              Page 8, Lines 215-223: Too much clinical details for a journal like "Pharmaceutics". Resume, if possible.

xi)             Page 8, Line 229: Remove "in" before “mitochondria”.

xii)                Page 9, Line 233: Drawbacks related with liver and lung uptake of 99mTc-Sestamibi and 99mTc-Tetrofosmim are not discussed. Alternative 99mTc tracers studied in the past few years to overcome these drawbacks should be mentioned (Please, check the recent review Molecules 202227(4), 1188; https://doi.org/10.3390/molecules27041188).

xiii)              Page 9, Lines 248 and 249: The English in the part “99mTc-labelled perfusion tracers SPECT” must be improved.

xiv)              Page 9, Line 273: Rewrite the part “..identify inflammation into infarct..”, as “into” is not the most appropriate preposition to be used.

xv)                Page 10, Line 308: Replace “patients” by “patient”.

xvi)              Page 10, Line 309: Correct “Figure 3” to “Figure 4”.

xvii)            Page 10, Line 309: I suggest that the authors should include a short discussion on the advantages and drawbacks of the different radiotracers for innervation imaging, and not only for 11C-HED.

xviii)           Page 10, Lines 316-317: it is mentioned in the legend  “first row 123I-MIBG, second row 99mTc-sestamibi” but this is not clear in the Figure. Please, identify better in the figure.

Author Response

The review paper on radiotracers for cardiac imaging, submitted by Carmela Nappi et al., is well-organized, well-written and covers the most important developments in this field. However, excessive clinical details are given in some parts of the manuscript. By contrast, a more comprehensive comparison of the performance of the different radiotracers is missing in some cases. I will be happy to recommend the publication of the manuscript after the authors address the following issues:

  1. Page 2, Line 37: Define the CABG abbreviation.

Response: CABG abbreviation has been eliminated and reported in extenso without abbreviation.

  1. Page 3, Line 77: Write “fluorodeoxyglucose” as 2-deoxy-2-fluoro-D-glucose” all along the text.

Response: Done.

  • Page 4, entries of Table 4: Replace “Half-life” by " Radionuclide Half-life”, to avoid confusion with the biological half-life of the radiopharmaceutical.

Response: Done.

  1. Page 4, Table 4: For 11C-acetate and 11C-HED, uniformize the value of positron energy (either 0.961 MeV or 0.96 MeV).

Response: Done.

  1. Page 5, Table 4: Rewrite “99mSestamibi” and “99mTetrofosmin” as “99mTc-Sestamibi” and “99mTc-Tetrofosmin”, respectively.

Response: Done.

  1. Page 5, Line 147: I believe that the authors wanted to mean “where” instead of “were”. Please, correct.

Response: Done.

  • Page 6 and 7, Figures 2 and 3:The resolution of the images needs to be improved.

Response: the resolution of the images has been increased.

  • Page 8, Line 184: Correct “11C” to “11C”.

Response: Done.

  1. Page 8, Line 190: Clarify how 15O-water can assess myocardium metabolism.

Response: We added a clarification regarding the possibility to evaluate metabolism by 15O-water.

  1. Page 8, Lines 215-223: Too much clinical details for a journal like "Pharmaceutics". Resume, if possible.

Response: This paragraph has been shortened.

  1. Page 8, Line 229: Remove "in" before “mitochondria”.

Response: Done.

  • Page 9, Line 233: Drawbacks related with liver and lung uptake of 99mTc-Sestamibi and 99mTc-Tetrofosmim are not discussed. Alternative 99mTc tracers studied in the past few years to overcome these drawbacks should be mentioned (Please, check the recent review Molecules 2022, 27(4), 1188).

Response: The paragraph has been modified as suggested and the reference has been added (#44).

  • Page 9, Lines 248 and 249: The English in the part “99mTc-labelled perfusion tracers SPECT” must be improved.

Response: Done.

  • Page 9, Line 273: Rewrite the part “identify inflammation into infarct”, as “into” is not the most appropriate preposition to be used.

Response: Done.

  1. Page 10, Line 308: Replace “patients” by “patient”.

Response: Done.

  • Page 10, Line 309: Correct “Figure 3” to “Figure 4”.

Response: Done.

  • Page 10, Line 309: I suggest that the authors should include a short discussion on the advantages and drawbacks of the different radiotracers for innervation imaging, and not only for 11C-HED.

Response: A short discussion on the advantages and drawbacks of the different radiotracers for innervation imaging has been provided and a reference has been added (#76).

  • Page 10, Lines 316-317: it is mentioned in the legend “first row 123I-MIBG, second row 99mTc-sestamibi” but this is not clear in the Figure. Please, identify better in the figure.

Response: A modified version of the figure has been provided in the manuscript.

Round 2

Reviewer 1 Report

I think that there is no problem with the contents, but the two points that caught my eye is the following.

1) I think it would be easier to see if the font for the footnotes in the figures was dropped by one. It's hard to tell the difference from the main text.

(Figure1, 2, 4,  Table 3, 5)

2) In Table 3, 

Na+/K+ is O.K.?? I think Na+/K+ is correct. 

Author Response

1) I think it would be easier to see if the font for the footnotes in the figures was dropped by one. It's hard to tell the difference from the main text.

(Figure1, 2, 4,  Table 3, 5)

Response: the font has been edited as required

2) In Table 3, Na+/K+ is O.K.?? I think Na+/K+ is correct. 

Response: the Table has been edited as required

Reviewer 3 Report

All concerns have been well addressed.

Author Response

Thank you

Reviewer 4 Report

The authors addressed all the raised issues and I am glad to recommend the publication of the manuscript in the current form.

Author Response

Thank you
